# Effect of Storage Time and Bacterial Strain on the Quality of Probiotic Goat’s Milk Using Different Types and Doses of Collagens

**DOI:** 10.3390/molecules28020657

**Published:** 2023-01-09

**Authors:** Kamil Szopa, Małgorzata Pawlos, Agata Znamirowska-Piotrowska

**Affiliations:** Department of Dairy Technology, Institute of Food Technology and Nutrition, University of Rzeszow, 35601 Rzeszów, Poland

**Keywords:** fermented goat’s milk, probiotic culture, bovine collagen, collagen hydrolysate

## Abstract

Recently, increasing attention has been focused on developing new products based on goat’s milk. Consumers positively perceive fermented goat’s milk products as health-promoting due to their nutritional value, digestibility, and potential source of probiotics. This study aimed to evaluate the possibility of using different doses of collagen and collagen hydrolysate in the production of probiotic goat’s milk fermented by four monocultures: *Lacticaseibacillus casei* 431^®^ *Lactobacillus acidophilus* LA- 5^®^, *Lacticaseibacillus paracasei* LP26, and *Lacticaseibicillus rhamnosus* Lr- 32^®^. A total of 20 experimental groups were prepared, including control groups (without additives), and due to the added probiotic (*Lacticaseibacillus casei, Lactobacillus acidophilus, Lacticaseibacillus paracasei*, and *Lacticaseibacillus rhamnosus*), various collagen doses (1.5% and 3.0%) and collagen types (hydrolysate and bovine collagen). Physicochemical, organoleptic, and microbiological characteristics were evaluated after 1 and 21 days of cold storage. The applied additives increased the acidity of the milk even before fermentation. However, milk with bovine collagen and hydrolysate had a higher pH value after fermentation than control milk. The study showed higher than 8 log cfu g^−1^ viability of probiotic bacteria in goat’s milk products during storage due to the proper pH, high buffering capacity, and rich nutrient content of goat’s milk. The best survival rate was shown for the *L. casei* strain after 21 days in milk with collagen protein hydrolysate. Moreover, collagen in milk fermented by *L. rhamnosus* decreased syneresis compared to its control counterpart. The addition of collagen, especially the hydrolysate, increased the gel hardness of the fermented milk. The collagen additives used in the milk, both in the form of hydrolysate and bovine collagen, caused a darkening of the color of the milk and increased the intensity of the milky-creamy and sweet taste.

## 1. Introduction

Milk-based products account for almost 43% of the functional food market and consist mainly of fermented products [1]. Probiotic fermented milk is gaining popularity due to its positive health-promoting benefits [2]. Fermented dairy products often provide carriers of probiotics, and it is well known that these products may enhance human health [3,4]. Moreover, probiotics are the main bioactive components of fermented dairy foods [5]. It is considered that the minimum number of viable probiotic cells in a product should be 10^6^ cfu g^−1^ at the end of the shelf life [6,7,8,9]. Whereas, according to the recommendations of the International Dairy Federation, probiotic products should contain at least 7 log cfu g^−1^ of lactic acid bacteria [10]. Consuming food containing about 10^8^ to 10^9^ cfu g^−1^ of probiotic micro-organisms daily provides health benefits to the consumer [8]. Various bioactive compounds, including peptides, exopolysaccharides, and short-chain fatty acids (SCFAs), could be obtained during the fermentation processes carried out by probiotic bacteria, thus enabling them to exert probiotic effects, such as improving intestinal health, preventing cancer, and improving immune response [2]. Selecting probiotic strains is based on their safety, functionality, and technological suitability. The probiotic potential is directly related to specific strains rather than the type or species of microorganism [11,12]. Popular probiotic strains include bacteria from the genus *Lactobacillus*, *Bifidobacterium*, and *Enterococcus* [13].

Consumers’ interest in goat’s milk and derived products is mainly related to its better digestibility for infants, the elderly, and patients with gastrointestinal disorders, and is also associated with lower allergenicity compared to cow’s milk [14,15,16]. Combining goat’s milk of high nutritional value with bacterial strains with documented probiotic properties is one of the possibilities for producing functional milk [5]. Technologically, the smaller size of the fat globules provides a smoother texture, and smaller amounts of the αs_1_ fraction of casein result in softer gel products, higher water-holding capacity, and lower viscosity [15,17]. In fermented goat’s milk technology, difficulties are met in obtaining an appropriately firm gel texture. This may be due to the low buffering capacity of goat’s milk and the poor concentration or absence of the αs1 fraction of casein [16,18]. These factors affect the rheological properties of goat coagulate, which is a semi-liquid product after fermentation.

In times of dynamic development of the functional food market and the necessity to provide an increasingly attractive variety, new products rich in prebiotics, polyphenols, minerals, vitamins, complete proteins, or essential polyunsaturated fatty acids, among others, are being developed [19]. Adding bioactive additives into fermented milk recipes results in products with higher nutritional value and bioavailability of ingredients. To increase the content of well-absorbed protein in milk, collagen or its hydrolysate can be applied through which milk with potential properties to prevent musculoskeletal diseases and injuries is obtained. The use of collagen hydrolysate provides the products with distinct therapeutic-preventive properties since collagen participates in the formation of connective tissue in the human body and thus can act as a biologically active additive with chondroprotective properties [19,20,21]. Oral intake of 10 g of collagen hydrolysate per day over three months reduces joint pain and improves bone fracture healing and joint function in patients [22,23].

There are many studies on the properties of probiotic fermented goat’s milk [15,16,24,25,26,27,28,29]. However, there is a lack of reports in the literature on the simultaneous effect of adding different types of collagen and probiotics on the properties of fermented goat’s milk during storage. The present study evaluated the potential of using different doses of collagen and collagen hydrolysate to produce probiotic fermented goat’s milk using four strains: *Lacticaseibacillus casei*, *Lacticaseibacillus paracasei*, *Lactobacillus acidophilus*, and *Lacticaseibacillus rhamnosus*. The effects of storage time, type, and dose of collagen and bacterial strains on fermented goat’s milk’s physicochemical, organoleptic, and microbiological properties after 1 and 21 days of refrigerated storage were studied.

## 2. Results and Discussion

Variability in the composition of goat’s milk affects the technological and sensory properties [30]. In order to determine the processing suitability of raw goat’s milk, the composition and physicochemical characteristics of goat’s milk were evaluated (Table 1). The obtained results are comparable with the findings of other studies [16,31,32].

### 2.1. Effect of Pasteurization on pH and Color of Milk with Collagen Addition

Milk acidity is an essential indicator of the quality of milk for the production of dairy products [33]. The results of studies of pH values and color parameters of goat’s milk after pasteurization of control samples and with the addition of collagen are shown in Table 2. It was found that both hydrolysate and bovine collagen increase the acidity of milk, which could result in a decrease in the thermal stability of milk proteins at higher doses. However, for this study, milk with 3.0% hydrolysate and with 3.0% bovine collagen held the pasteurization temperature (85 °C, 10 min), and no protein denaturation was observed. Collagen was added to milk with a pH value of 6.8. Under these conditions, at pH values below isoelectric point, there is an increase in hydrogen ions and protein particles are positively charged, causing repulsion between them [34,35]. Therefore, the pH value is lower in milk with collagen than in control milk. Color parameters were also monitored in the goat’s milk after pasteurization. Most importantly, the addition of collagen did not significantly change the color components of goat’s milk (Table 2). Compared to control milk, there was only a trend of reduced green and yellow color intensity in milk with hydrolysate and bovine collagen.

### 2.2. Acidity of Probiotic Goat’s Milk with Collagen

The application of milk from different animal species could influence the acidification rate by lactic acid bacteria, including probiotic bacteria. Therefore, some replicate better in goat’s milk than in milk from other animal species [36]. Some authors indicated that the higher fermentation activity of lactic acid bacteria in goat’s milk is due to its specific composition and structure [37]. The buffering capacity of goat’s milk is lower than cow’s and sheep’s milk. During the manufacture of yogurt from goat’s milk, a pH of 4.6 to 4.7 is achieved after 2 h 45 min; in cow’s milk, after 3 h 30 min; and in sheep’s milk, after 5 h 30 min. The faster pH changes in goat’s milk are mainly due to its lower casein content, while β-casein is the least phosphorylated. Moreover, the higher non-protein nitrogen content, higher vitamin concentration, higher amount of some minerals and short-chain fatty acids, and easier protein digestibility in goat’s milk may affect the faster increase in acidity [38]. Additionally, our study observed different pH values on day 1 of storage in control samples fermented by strains: *Lacticaseibacillus casei, Lactobacillus acidophilus, Lacticaseibacillus paracasei,* and *Lacticaseibacillus rhamnosus* (Figure 1, Figure 2, Figure 3 and Figure 4). For all control samples, the lowest pH value was determined in LA control milk fermented by *L. acidophilus* (pH = 4.33) and the highest in LP samples with *L. paracasei* (pH = 4.54) on day 1 of storage. The acidity of the obtained milk is also affected by the type of collagen used. The addition of both collagen hydrolysate and bovine collagen reduced the pH value of the milk even before fermentation. However, adding hydrolysate was significantly more influenced by increasing the pH value than bovine collagen. These differences were statistically significant (*p* ≤ 0.05) at 1.5% and 3.0% collagen dosage in all fermented milk samples compared to their control counterparts. It was found that the addition of 3.0% hydrolysate and bovine collagen results in an increase in pH values by 0.42 in LA3.0W samples and 0.54 in LA3.0H samples of fermented milk. Moreover, in Goto’s [39] study, the addition of 1.5% bovine collagen resulted in a 0.14 reduction in pH values, and the addition of 3.0% collagen resulted in a 0.23 decrease in pH values.

After 21 days of storage, all samples with collagen showed higher pH values than their control counterparts. In fermented milk, after 21 days of storage, adding 1.5%, bovine collagen increased the pH value from 0.03 in LR1.5W to 0.17 in LA1.5W. However, adding 3% collagen increased the pH value to 0.17 in LR3.0W and 0.31 in LA3.0W. After 21 days of storage, the addition of 1.5% hydrolysate increased the pH value compared to controls from 0.11 in LP1.5H to 0.28 in LA1.5H. Increasing the dose resulted in higher pH values, from 0.20 in LP3.0H and LC3.0H to 0.39 in LA3.0H. Studies by Goto [39] indicated a slow reduction in pH in milk containing 3.06% and 5.1% collagen, indicating a delayed fermentation phenomenon caused by the addition of collagen. Similar results were obtained in milk with different types of collagen, i.e., fish and pork [39].

A study by Znamirowska et al. [40] of cow’s milk fermented by *L. rhamnosus* similarly showed higher pH values in milk with collagen hydrolysate compared to control samples during 21-day storage. A study by Dimitrellou et al. [15] found that probiotic yogurts obtained from goat’s milk at 21 days of refrigerated storage had higher acidity (pH = 3.88) compared to those obtained from cow’s milk (pH = 3.98). Probiotic sheep’s milk with collagen hydrolysate obtained in the study by Szopa et al. [9] was also characterized by higher pH values than the control samples.

The results in Table 3, Table 4, Table 5 and Table 6 indicate that the lactic acid content depended on the fermentation strain. On day 1 of storage, the highest lactic acid content of all control groups was found in LA milk fermented by *L. acidophilus* (Table 4). In contrast, the lowest was found in LP samples fermented by *L. paracasei* (Table 5). The lactic acid content of the samples with collagen protein hydrolysate and bovine collagen was higher on the 1st and 21st day of the study compared to the control groups of milk fermented by *L. casei, L. paracasei,* and *L. rhamnosus*.

On day 1 of storage of samples fermented by *L. paracasei*, the addition of bovine collagen was found to increase the lactic acid content by 0.1 g L^−1^ compared to the control sample while increasing the dose of bovine collagen from 1.5% to 3.0% did not significantly affect the lactic acid content of fermented milk. However, the hydrolysate addition to milk fermented by *L. paracasei* increased lactic acid content by 0.08 g L^−1^ but only at the 3.0% hydrolysate dose. At a dose of 1.5% hydrolysate, the lactic acid content was the same as in the control sample on day 1 of storage (Table 5). However, the concentration of lactic acid in samples with collagen hydrolysate (LA1.5H and LA3.0H) fermented by *L. acidophilus* (Table 4) was lower than in LA control milk and milk with bovine collagen (LA1.5W and LA3.0W).

After 21 days of storage, the highest lactic acid content was determined in samples fermented by *L. casei* (Table 3) with a 3.0% addition of hydrolysate LC3.0H and samples with bovine collagen LC3.0W, where it was determined, respectively, 1.35 g L^−1^ and 1.34 g L^−1^. The highest concentration of lactic acid after 21 days of storage was found in goat’s milk fermented by *L. casei*. In this case, adding bovine collagen at 1.5% and 3.0% increased the acid content by 0.04 g L^−1^ and 0.12 g L^−1^, respectively. Additionally, the addition of hydrolysate increased the lactic acid concentration in the LC1.5H sample by 0.08 g L^−1^ and the LC3.0H sample by 0.13 g L^−1^ after 21 days of storage. Moreover, extending the storage time to 21 days resulted in the highest increase in lactic acid concentration in samples fermented by *L. casei* compared to the other fermented milk samples. A similar correlation was shown by Znamirowska et al. [40], and according to Kavaz and Bakirci [41], the amount of lactic acid in probiotic yogurts increases with storage time. In a study performed by Shori et al. [42], the addition of fish collagen increased the initial titratable acidity (TA%) by about 0.2% lactic acid equivalent. In yogurts with fish collagen, there was an increase in TTA (total titratable acidity) throughout refrigerated storage compared to the control sample [43].

### 2.3. Syneresis of Probiotic Goat’s Milk with Collagen

Syneresis is considered one of the most apparent defects in fermented milk, resulting from leakage of yellow-green whey on the surface of fermented milk [16]. Syneresis occurs due to shrinkage of the protein gel, which leads to the separation of whey and curd. The process of whey separation is related to the firmness and stability of the protein network and other factors, such as the type of milk, low pH, high acidity, the type and intensity of heat treatment, storage time, and the type of additives and stabilizers used [40,44,45]. Acid gel syneresis is also significantly affected by the type of starter culture used for milk fermentation due to the metabolites produced, including exopolysaccharides [46]. 

In our study, on day 1 of storage, in most cases, adding a hydrolysate at 3.0% increased syneresis from 0.2% to 9.0%, depending on the type of strain used (Table 3, Table 4, Table 5 and Table 6). LA3.0H fermented milk showed a 2.42% reduction in syneresis compared to the control; however, the reported difference was not statistically significant (Table 4). Conversely, the most significant reduction in syneresis compared to the control sample (by 2.58%) was found in milk fermented by *L. rhamnosus* with the addition of 1.5% hydrolysate (Table 6).

After 21 days of storage, all samples of milk fermented by *L. rhamnosus* showed an increase in syneresis of 1.83–4.53% compared to day 1 of storage. Furthermore, significantly lower syneresis was found in samples with bovine collagen LR1.5W and LR3.0W compared to the control sample LR fermented by *L. rhamnosus* both on day 1 and day 21 of storage (Table 6). However, the milk with collagen fermented by *L. casei* (Table 3) had lower syneresis than the control milk samples on day 21 of refrigerated storage. Trends toward lower syneresis with increased storage time were also found in samples fermented by *L. paracasei* (Table 5).

Gomes et al. [44] showed weaker curd and reduced whey leakage in beverages made from goat’s milk compared to cow’s milk. This was related to the composition and microstructure of goat’s milk, which has smaller casein micelles than cow’s milk, resulting in a protein network with smaller pores, higher density, and lower water-holding capacity. Gerhardt et al. [47] indicated that adding collagen hydrolysate above 1.0% reduced the syneresis intensity of fermented milk, improving its stability. In our study, adding 1.5% hydrolysate reduced syneresis only in milk fermented by *L. rhamnosus*, as syneresis was determined by the milk’s pH value (r = −0.6040) and lactic acid content (r = 0.6671).

One of the most commonly used additives along with fruit loading is bovine gelatin, as it has a high ability to give products increased firmness and lower syneresis [35]. Consequently, goat’s milk yogurts with this additive have a less firm gel structure than their cow’s milk counterparts [48,49,50,51,52].

### 2.4. Color of Probiotic Goat’s Milk with Collagen

Many studies reported that milk additives affect color parameters. In our study, adding hydrolysate and bovine collagen caused color darkening during 21 days of refrigerated storage (Figure 5, Figure 6, Figure 7 and Figure 8).

The most significant color darkening on day 1 of storage was found in the LP3.0W sample fermented by *L. paracasei* (Figure 7). It was also found that L* color lightness tends to decrease with increasing storage time in all samples. Additionally, in the study by Rigoto et al. [53], the lightness and color angle values did not change for all milk samples with collagen during the storage period.

In our study, all milk samples were characterized by the proportion of green color (−a*) increasing in intensity with increasing storage time. Moreover, a significant negative correlation was shown between syneresis and the parameter b* (r = −0.6815) and color angle (r = −0.6691). The proportion of yellow color depended on the probiotic strain used and the concentration and type of collagen used. There was observed a reduction in the intensity of yellow color in milk fermented by *L. acidophilus* and *L. paracasei* (Figure 6 and Figure 7) on day 1 of storage with the addition of both collagen and hydrolysate and in milk with a 3% addition of collagen hydrolysate fermented by *L. rhamnosus* (Figure 8) compared to control milk. In milk fermented by *L. casei*, the addition of collagen and hydrolysate in two doses increased the proportion of yellow color on day 1 as well as day 21 of refrigerated storage (Figure 5). However, milk fermented by *L. rhamnosus* (Figure 8) on both day 1 and day 21 of storage showed a higher proportion of yellow color for samples with collagen. The other color components, C and h^0^, were within the range: C: 7.33—10.03, h^0^: 98.68—104.00.

Milk fermented by *L. acidophilus* and *L. rhamnosus* obtained by Szajnar et al. [13] was also characterized by light color and the proportion of green (−a*) and yellow (+b*) color. The study of Lasik and Pikul [54] observed that during the fermentation process in all samples, the parameter a* decreases and the parameter b* increases. This correlation is due to the acidification and coagulation process. Color changes also occurred during the non-enzymatic reaction starting with the binding of the aldehyde group of lactose to the amino group of milk proteins. The incubation temperature also affected the increase in the color parameter b*. Thus, the higher the incubation temperature, the higher the b* parameter [54].

### 2.5. Texture Profile of Probiotic Goat’s Milk with Collagen

Texture characterizes the product’s physical properties, such as hardness, adhesiveness, and springiness, and is considered among the essential quality characteristics of fermented dairy products. The texture properties of fermented milk are related to the type of milk, heat treatment, type of starter cultures, dry matter content, and the type of additives used [9,55]. Fermented goat’s milk shows a soft gel and a high ability to spread the curd compared to cow’s milk [55]. The rheological properties and texture of fermented goat’s milk could be improved by adding strains that produce exopolysaccharides (EPS) during incubation.

Goat’s milk gels have a lower solidity and a softer texture than cow’s milk gels. These properties are directly related to the smaller diameter of casein micelles, a lower degree of hydration, lower mineralization and casein content of milk, especially the αs_1_ casein fraction, and lower non-protein nitrogen diameter in goat’s milk compared to cow’s milk [16,55,56]. Smaller amounts or the lack of αs_1_ casein in goat’s milk results in a softer gel with lower viscosity [57,58]. In order to obtain the right curd consistency in fermented goat’s milk, it is necessary to increase the solids-non-fat content through the use of whey or milk protein concentrates or isolates, as well as caseinates, stabilizers, pectins, starches, and alginates, and even the addition of lactic acid bacteria (LAB) as exopolysaccharide generators [59,60,61]. Moreover, adding collagen to milk before fermentation could determine the physicochemical characteristics and hardness of the probiotic milk gel [9,40]. Adding proteins to goat’s milk is recommended to obtain a gel network that can be less susceptible to breaking and highly capable of immobilizing the aqueous phase in the matrix. Gel firmness increases with increasing protein levels [57,62]. Moreover, probiotic bacteria could determine the texture of fermented milk due to differences in the number of organic acids and exopolysaccharides produced [57,63,64].

This was also confirmed in our study, where the lowest gel hardness was found in the LA control sample fermented by *L. acidophillus* on both day 1 and day 21 of the experiment (Table 4). The hardness in the remaining control samples was 0.1–0.2 N higher. Moreover, it showed a significant negative correlation between hardness and pH (r > −0.6) on days 1 and 21 of storage. However, the hardest was the LR3.0H milk gel with 3.0% hydrolysate fermented by *L. rhamnosus* on day 1 of storage (Table 6). The hardness results in Table 3, Table 4, Table 5 and Table 6 showed a tendency to increase hardness after adding collagen, especially hydrolysate, compared to control samples on days 1 and 21 of storage. However, in most cases, the differences are not significant. Additionally, the springiness of the goat’s milk gel did not change significantly after the addition of collagen. In contrast, the 1.5% addition of bovine collagen reduced the cohesiveness of fermented milk compared to the control counterparts. However, statistically significant differences were shown only for LP1.5W samples after 21 days of storage. Additionally, in the Luo et al. [65] study, gel cohesiveness was not affected by the addition of 0.1% or 0.2% gelatin but increased significantly at higher gelatin concentrations. In a study by Pang et al. [66], the cohesiveness of milk gel samples also decreased significantly as the gelatin concentration increased to 1%, which could be related to steric interference by gelatin addition. These authors, studying the microstructure, observed that low concentrations of gelatin did not significantly change the gel matrix of milk [66]. An analysis of variance confirmed that the texture components (hardness, cohesiveness, and springiness) were most influenced by the type of probiotic strain used, the type of collagen, and the interaction of these two factors. In a study by Szopa et al. [9] in sheep’s milk fermented by *L. casei*, a reduction in milk cohesiveness was observed in samples with 1.5 and 3.0% collagen hydrolysate addition compared to the control sample throughout the storage period.

### 2.6. Viability of Probiotic Bacteria in Fermented Goat’s Milk with Collagen

Probiotic bacteria could significantly impact the quality of fermented milk and the formation of metabolites during fermentation and storage [2]. The viability of *lactobacilli* in fermented milk depends on several factors, including fermentation time and temperature, product storage conditions, acidity, dry matter and carbohydrate content, bacterial access to nutrients, the presence of oxygen, and the type, species, and strain of lactic acid bacteria used for fermentation [67]. In Mituniewicz-Malek et al.’s [67] study, during the whole period of 21-day refrigerated storage, the population of probiotic strains used in the production of fermented goat’s milk did not change significantly, and all experimental beverages were characterized by a normative number of viable cells (at least 6 log cfu g^−1^). Minervini et al. [68] found that in fermented goat’s milk immediately after fermentation, the number of probiotic strains reached 7–8 log cfu g^−1^, while after 45 days of refrigerated storage of fermented milk, it was significantly reduced, and for *L. casei*, it was 7.0 log cfu g^−1^ [68]. Yerlikaya et al. [64] showed the highest viability for *L. acidophilus* strains and the lowest for *L. casei* during 30 days of storage. Kim et al. [69] showed that collagen has a protective effect and increases the viability of probiotic strains and enhances bacterial stability, survival in the gastrointestinal tract, and heat resistance.

In our study, the acceptable survival rate of probiotic bacteria (>8 log cfu g^−1^) could be attributed to the properties of probiotic strains. The best survival rate after 21 days of cold storage compared to day 1 was found in milk with 3.0% added collagen hydrolysate fermented by *L. casei* (LC3.0H). The survival rate was 103.66%. This could be explained by considering the nutritional requirements of this strain, which indicate a high consumption of amino acids. Zhang et al. [70] added various nutrients to the fermentation of milk with *L. casei*, such as asparagine, cysteine, proline, glycine, glutamic acid, tyrosine, guanine, Ca-pantothenate, pyridoxine, Mn^2+^, and Mg^2+^, reducing the fermentation time and increasing the survival rate of the strain.

The analysis of the results in Table 7 shows that adding collagen hydrolysate to probiotic milk had no significant effect on the viability of *L. rhamnosus* and *L. paracasei* on day 1 and day 21 of storage. As Morita et al. [71] demonstrated, *L. rhamnosus* can hydrolyze milk proteins to obtain the appropriate amino acids and synthesize other nutrients primarily early in the fermentation process (0–8 h). *L. rhamnosus* can synthesize nutrients needed later in fermentation, but this ability is poor [72]. Thus, the addition of collagen was also found to be insignificant.

However, milk with hydrolysate LC1.5H, LC3.0H, and LA3.0H showed significantly higher counts of live probiotic cells after 21 days than their control counterparts (Table 7). Meng et al. [73] analyzed the nutritional requirements of *L. acidophilus* LA-5, including the consumption standards of amino acids, purines, pyrimidines, vitamins, and metal ions. The nutrients required by *L. acidophilus* LA-5 were asparagine, aspartic acid, cysteine, leucine, methionine, riboflavin, guanine, uracil, and Mn ^2+^, and when added to milk, the fermentation time of milk fermented by *L. acidophillus* LA-5 was reduced by 9 h, with a high number of viable cells that persisted during storage of the fermented milk supplemented with nutrients compared to the control. Therefore, adding an amino acid-rich hydrolysate to goat’s milk before fermentation stimulates this strain’s survival rate.

Significantly, all values (8.2–9.3 log cfu g^−1^) corresponded to the recommended minimum daily intake of live probiotic cells per serving of ready-to-eat product [74]. These results demonstrate the functional potential of final fermented milk, in which the probiotic content could be considered sufficient to provide benefits to consumers and could compensate for any possible limitations caused by passage through the gastrointestinal tract [75].

### 2.7. Organoleptic Evaluation of Probiotic Goat’s Milk with Collagen

The most important problem related to applying goat’s milk to several dairy products is its goaty taste and odor [44]. The specific odor of goat’s milk is linked to the composition of fatty acids. The content of caproic, caprylic, and capric acids in goat’s milk is slightly higher than in cow’s milk. These acids are present in large quantities in the fat of goat’s milk and are released from the membranes of the fat globules by lipases in case of improper milking and processing. During fermentation, goat’s milk changes taste and odor [76,77]. It was observed that in fermented milk derived from the milk of other animals (goat, sheep), the volatile compounds’ profile differs, giving them specific sensory properties. In addition to the fermentation process, many of the milk’s native compounds are metabolized using different bacterial cultures, yielding different metabolites that contribute to specific aromatic notes.

The results of the evaluation of goat’s milk fermented by four monocultures, including control and with collagen, are shown in Figure 9, Figure 10, Figure 11 and Figure 12. Among the control samples on day 1 of storage, LA (Figure 10) milk was the sourest, and LP (Figure 11) milk was the least, which were also the sweetest. Fermented goat’s milk with *L. acidophilus* and *L. casei* tested by Mituniewicz-Malek [67] was characterized by a more noticeable sour and goaty aftertaste, minimal whey leakage, and looser texture.

In this study, the panelists found that adding collagen increased the intensity of the milky-creamy and sweet taste. These effects were especially noticeable after 21 days of storage. The intensity of off-taste and additive taste depended on the bacterial strain and storage time. The most intense off-taste and odor were found in LA milk while adding bovine collagen and hydrolysate reduced their perceptibility.

The goat’s fermented milk samples were characterized by a mildly sour odor (Figure 9, Figure 10, Figure 11 and Figure 12). Acetaldehyde is an important compound that gives milk its fermentation odor. In fermented beverages made from goat’s milk, compared to cow’s milk, it is difficult to have a high content of acetaldehyde, which is conditioned by the more than twenty-fold higher content of glycine, which has an inhibitory effect on threonine aldolase, converting threonine into acetaldehyde and glycine [67]. According to Gao et al. [78], milk fermented by *L. casei* has a higher content of acetic acid, lactic acid, butyric acid, caproic acid, acetoin, 2-butanone, and 2-ethyl-1-hexanol, which may result in higher product acceptance. Adding whey protein products to milk increased precursors in the acetaldehyde formation pathway. Whey protein products are rich in threonine and valine AA compared to caseinates, which might be why the high concentration of acetaldehyde in yogurts is fortified with whey protein-based additives [57].

## 3. Materials and Methods

### 3.1. Materials

The material for producing probiotic fermented milk was raw goat’s milk collected in June 2022 from an organic farm in the Podkarpacie region (Zabratówka, Poland) from goats of mixed breeds. Milk was pre-filtered to remove dirt and foreign particles. The methods of milk analysis are presented in Section 3.3.

Two types of collagens were used as additives: 100% collagen protein hydrolysate (Vitagel-Collagen, Superior, Dobre Miasto, Poland) and bovine collagen—100% natural (FH Kol-Pol, Dębica, Poland). For the fermentation of milk, four strains of probiotic bacteria were used: *Lacticaseibacillus casei* 431^®^ (Chr. Hansen, Hoersholm, Denmark), *Lactobacillus acidophilus* LA- 5^®^ (Chr. Hansen, Hoersholm, Denmark), *Lacticaseibacillus paracasei* LP26 (DELVO^®^ PRO, DSM, Australia), and *Lacticaseibicillus rhamnosus* Lr- 32^®^ (Danisco, DuPont, Copenhagen, Denmark).

IBCm Bacto Kit 500 and IBCm SCC Kit reagents for determining TBC (total bacterial count) and SCC (somatic cell count) were purchased from Bentley Instruments Inc. (Chaska, MN, USA). MRS agars (De Man, Rogosa and Sharpe) and peptone water came from Biocorp (Warszawa, Poland). Sodium hydroxide and phenolphthalein were purchased from Chempur (Piekary Śląskie, Poland).

All of the reagents used were of analytical reagent grade.

### 3.2. Fermented Milk Manufacture

Raw goat’s milk was pasteurized at 85 °C for 30 min [79]. The chilled milk was divided into 20 groups according to the probiotic strain added (*Lacticaseibacillus casei, Lactobacillus acidophilus, Lacticaseibacillus paracasei,* and *Lacticaseibacillus rhamnosus*) and different types of collagen (hydrolysate and bovine collagen) and dosages of collagen (1.5% and 3.0%). Five groups of milk were inoculated with one of four pre-activated (5% starter cultures) bacterial monocultures according to the method of Szajnar et al. [13]. The first group was without collagen addition (control), the second group contained 1.5% collagen, the third group contained 3.0% collagen, the fourth group contained 1.5% collagen hydrolysate, and 3.0% collagen hydrolysate was added to the fifth group. The milk–collagen mixture was heated to 60 °C, homogenized at 20 MPa (Nuoni GJJ-0.06/40, Shanghai, China), and re-pasteurized according to the method of Ramasubramanian et al. [80] (EC) and Commission Regulation (EC) No. 1662/2006 [81] with modifications (85 °C, 10 min). After heat treatment, the milk samples were cooled to inoculation temperature (37 ± 1 °C). A total of 20 batches of milk were obtained, according to Figure 1.

Each sample was mixed, transferred into 100 mL plastic containers, and fermented in an incubator at 37 ± 1 °C in order to obtain a pH value of 4.6 ± 0.2 (12–15 h). After this period, the fermented milk was cooled to 5 °C (ILW 115 Refrigerated Incubator, POL-EKO Aparatura, Wodzisław Śląski, Poland). The evaluation of fermented milk was carried out on days 1 and 21 of cold storage. The experiment was repeated three times, and all analyses were performed in five replicates each time.

### 3.3. Methods of Analyses

#### 3.3.1. Determination of Freezing Point and Chemical Composition of Raw Goat’s Milk

Freezing point and chemical composition of goat’s milk were determined using milk and milk product analyzer Bentley B-150 (Bentley Instruments Inc., Chaska, MN, USA).

The density of milk was measured at a temperature of 20 °C, according to the method used by Ratu et al. [82].

#### 3.3.2. Microbiological and Cytological Quality of Raw Goat’s Milk

Total bacterial count (TBC) and somatic cell count (SCC) were performed using a semi-automated Bacto Count IBC M/SCC (Bentley Instruments Inc., Chaska, MN, USA).

#### 3.3.3. Titratable Acidity Expressed as Lactic Acid Content and pH

Lactic acid content was determined according to Jemaa et al. [83]. Fermented milk samples were titrated with 0.1 M NaOH in the presence of phenolphthalein as an indicator and lactic acid content was expressed as g lactic acid L^−1^.

The pH value in raw goat’s milk, milk with collagen and collagen hydrolysate addition and after fermentation, was determined by pH-meter FiveEasy (Mettler Toledo, Greifensee, Switzerland) using InLab^®®^Solids Pro-ISM electrode (Mettler Toledo, Greifensee, Switzerland) [84].

#### 3.3.4. Color Evaluation

The color of raw goat’s milk, milk with collagen and collagen hydrolysate addition and after fermentation, was determined by a colorimeter (Precision Colorimeter, Model NR 145, Shenzhen, China) using the CIE L*a*b* system. The following parameters were determined: L*—as lightness (from 0—black to 100—white), a*—as color from red (+) to green (−), b*—as color from yellow (+) to blue (−), C—as color purity and intensity, and h^0^—as color hue [85]. Before measurement, the device was calibrated on a white reference standard [86].

#### 3.3.5. Syneresis

Determination of syneresis was performed by the centrifuge method using the laboratory refrigeration centrifuge LMC-4200R (Biosan SIA, Riga, Latvia) according to Santillan-Urquiza et al.’s [87] method with modifications: 10 g of product was transferred into a 50 mL plastic tube and centrifuged at 4000 rpm for 10 min. Syneresis percentage was expressed as the amount of whey released compared to the initial sample weight and multiplied for 100.

#### 3.3.6. Texture Profile Analysis

Texture profile analysis (TPA) was determined by an instrumental method using CT3 Texture Analyzer with Texture Pro CT software (Brookfield AMETEK, Berwyn, PA, USA), according to Znamirowska et al. [16]. The cylindrical dimensions of the sample were as follows: 66.00 mm × 33.86 mm. The sample temperature was 8 °C. The settings used were trigger load 0.1 N; test speed 1 mm/s; TABTKIT table; probe TA3/100 (acrylic cylinder—diameter 35 mm); and test termination distance: 15 mm. The parameters measured were hardness [N], cohesiveness, and springiness [mm].

#### 3.3.7. Microbiological Analysis

The number of probiotic bacteria (*Lacticaseibacillus casei* 431, *Lactobacillus acidophilus* LA-5, *Lacticaseibacillus paracasei* Lpc-37, and *Lacticaseibacillus rhamnosus* Lr-32) was determined by plate method using MRS agar according to the method of Znamirowska et al. [40] and Lima et al. [86]. Incubation was conducted under anaerobic conditions at 37 °C for 72 h in a vacuum desiccator and GENbox anaerator (Biomerieux, Warsaw, Poland). After incubation, colonies were counted using a colony counter (TYP J-3, Chemland, Stagard Szczeciński, Poland). The number of viable bacterial cells was expressed as log cfu g^−1^. From the obtained results, the percentage survival rate of probiotic bacteria was calculated. The survival rate of probiotic bacteria (%) was determined by the number of viable colonies of probiotic bacteria in the fermented milk on day 21 of storage relative to the sample from day 1 of storage, according to Equation (1):(1)$$\mathrm{Survival}\;\mathrm{rate}\;\mathrm{of}\;\mathrm{probiotic}\;\mathrm{bacteria}\;(\%)=\frac{\mathrm{Viable}\;\mathrm{counts}\;\mathrm{of}\;\mathrm{probiotic}\;\mathrm{bacteria}\;\mathrm{in}\;\mathrm{fermented}\;\mathrm{milk}\;\mathrm{on}\;\mathrm{day}\;21\;\mathrm{of}\;\mathrm{storage}}{\mathrm{Viable}\;\mathrm{counts}\;\mathrm{of}\;\mathrm{probiotic}\;\mathrm{bacteria}\;\mathrm{in}\;\mathrm{fermented}\;\mathrm{milk}\;\mathrm{on}\;\mathrm{day}\;1\;\mathrm{of}\;\mathrm{storage}}\;\times \;100$$

#### 3.3.8. Organoleptic Evaluation

The organoleptic evaluation was carried out by a trained panel for probiotic fermented milk enriched with collagen and collagen hydrolysate at 1 and 21 days of cold storage. The parameters were evaluated on a 9-point scale (from 1 = undetectable to 9 = very intense). The following parameters were evaluated: texture, color, smoothness, presence of milky-creamy taste, sour taste, sweet taste, additives taste and off-taste (bitter, metallic), and the presence of sour odor, additives odor, and off-odor [1,88].

The definitions of the attributes in the descriptive organoleptic evaluation of fermented milk [10] are presented in Table 8.

#### 3.3.9. Statistical Analysis

From the obtained results, the mean, standard deviation, and Pearson’s correlation coefficient were calculated using Statistica v. 13.1 software (StatSoft, Tulsa, OK, USA). ANOVA analysis was conducted to investigate the overall effect of collagen type and dose, storage time (days), and type of bacteria on the properties of probiotic fermented goat’s milk. The significance of the differences between means was estimated by the Tukey test (*p* < 0.05).

## 4. Conclusions

This study confirmed the possibility of using bovine collagen and hydrolysate to produce probiotic goat’s milk. In all groups of fermented milk, the number of colony-forming viable probiotic bacterial cells obtained throughout the storage period was higher than 8 log cfu g^−1^. Adding hydrolysate and bovine collagen significantly improved bacterial survival on the 21st day of storage in milk fermented by *L. casei* and *L. acidophilus* compared to their control counterparts. These results demonstrate the functional potential of shelf-stable fermented milk, in which the probiotic content could be considered sufficient to provide health benefits to consumers. Used additives increased the acidity of the milk even before fermentation. However, after fermentation of the milk with bovine collagen and hydrolysate were characterized by higher pH values compared to control milk. This is essential information for dairy technologists relating to fermentation timing to determine the product’s specific pH and the number of viable probiotic bacteria cells in the product. The intensity of off-taste and the additive was based on the bacterial strain and storage time. The most important problem concerning the use of goat’s milk for dairy products is its goaty taste and odor. However, the addition of collagen did not increase the goat taste perception but intensified the milky-creamy and sweet taste and caused a darkening of the color. The taste and color of fermented goat’s milk could be easily improved by using fruit or chocolate flavorings. Moreover, collagen in milk fermented by *L. rhamnosus* decreased syneresis compared to its control counterpart. The addition of collagen, especially the hydrolysate, increased the gel hardness of the fermented milk. The collagen additives used in the milk, both in the form of hydrolysate and bovine collagen, caused a darkening of the color of the milk and increased the intensity of the milky-creamy and sweet taste.

Overall, our results indicate that probiotic goat milk with collagen has high commercial potential due to its nutritional and functional composition. The supplementation of fermented goat’s milk with collagen could be a new direction for broadening the assortment of goat products. However, it still requires improvements in technological parameters, mainly fermentation with different probiotic strains.

Nevertheless, further research is necessary to adequately clarify the impact and correlation between the content of different types of collagen and hydrolysates (not just bovine) and the quality of fermented milk made from the milk of different animal species. In particular, research is required on the survival rate of different strains of probiotic bacteria delivered to the human body in the form of fermented milk with collagen determined by simulated in vitro digestion. Moreover, the release rate of nutrients could also be studied, which is essential for products intended for consumers with special needs.

## Data Availability

The original data presented in the study are included in the article, and further inquiries can be directed to the corresponding author.

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
