# Peer review of "Effect of Storage Time and Bacterial Strain on the Quality of Probiotic Goat’s Milk Using Different Types and Doses of Collagens"

_molecules, 2023, doi:10.3390/molecules28020657_

Round 1
Reviewer 1 Report
I have revised the manuscript entitled “Effect of Storage Time and Bacterial Strain on the Quality of Probiotic Goat’s Milk with Different Types of Collagen”
This manuscript is very interesting, since it is intended to provide data for the development of a new product based on goat´s milk, and added with collagen. The article is very well detailed in methodology section, but the results contain a lot of tables and the information is very extensive. The discussion section is within results section but in some parts are confuse. I suggest major revision.
Introduction: Is well explained.
Line 30: you state that functional products represent 43%. ¿Of what, of the world production? ¿What did you mean? Please clarify.
Methodology: In general, is very descriptive and understandable. Also refers modern methods of high quality and reliability.
Line 416: Please clarify the acronymus of TBC and SCC.
Line 417: MRS did you mean De Mann Rogosa Sharpe? Please clarify and add the brand of the medium.
Results and discussion: The results are very long and contain many tables (table 1-5) and it seems very confusing. It is suggested to try to remove some tables and add representative graph.
Tables 1-5 are repetitive and very extensive. Is there the possibility of removing some and making a representative graph for some parameters?
I suggest adding subtitles to highlight each result with its respective discussion, and avoid confusion.
Author Response
Reviewer 1
The authors would like to thank for valuable comments and suggestions. The text of the manuscript has been revised as recommended.
I have revised the manuscript entitled “Effect of Storage Time and Bacterial Strain on the Quality of Probiotic Goat’s Milk with Different Types of Collagen”
This manuscript is very interesting, since it is intended to provide data for the development of a new product based on goat´s milk, and added with collagen. The article is very well detailed in methodology section, but the results contain a lot of tables and the information is very extensive. The discussion section is within results section but in some parts are confuse. I suggest major revision.
Introduction: Is well explained.
Line 30: you state that functional products represent 43%. ¿Of what, of the world production? ¿What did you mean? Please clarify.
The sentence was corrected:
L32-33: Milk-based products account for almost 43% of the functional food market and consist mainly of fermented products.
Methodology: In general, is very descriptive and understandable. Also refers modern methods of high quality and reliability.
Line 416: Please clarify the acronymus of TBC and SCC.
L576-577: The meaning of the acronymus TBC and SCC was explained:
“IBCm Bacto Kit 500 and IBCm SCC Kit reagents for determining TBC (total bacterial count) and SCC (somatic cell count) were purchased from Bentley Instruments Inc. (Chaska, MN, USA).”
Line 417: MRS did you mean De Mann Rogosa Sharpe? Please clarify and add the brand of the medium.
L578: The meaning of MRS abbreviation was explained:
„MRS agars (De Man, Rogosa and Sharpe) and peptone water came from Biocorp (Warszawa, Poland). Sodium hydroxide and phenolphthalein were purchased from Chempur (Piekary ÅšlÄ…skie, Poland).”
Results and discussion: The results are very long and contain many tables (table 1-5) and it seems very confusing. It is suggested to try to remove some tables and add representative graph.
Tables 1-5 are repetitive and very extensive. Is there the possibility of removing some and making a representative graph for some parameters?
Some of the results (pH value and color of probiotic fermented milk) in Table 3-6 are removed and presented in Figures 1-8:
L161-193: Figures 1-4
L325-360: Figures 5-8
Figure 1. Effect of collagen addition on pH parameters of milk fermented by L. casei after 1 and 21 days of cold storage.
Mean ± standard deviation; n = 20; a–d— Mean values denoted by different lowercase letters indicate statistically significantly at p ≤ 0.05 depending on the collagen type and dose; A,B—Mean values denoted by different capital letters indicate statistically significantly at p ≤ 0.05 depending on the day of storage; LC: control milk; LC1.5W: milk with 1.5% collagen; LC3.0W: milk with 3.0% collagen; LC1.5H: milk with 1.5% collagen protein hydrolysate; LC3.0H: milk with 3.0% collagen protein hydrolysate.
Figure 2. Effect of collagen addition on pH parameters of milk fermented by L. acidophilus after 1 and 21 days of cold storage.
Mean ± standard deviation; n = 20; a–d— Mean values denoted by different lowercase letters indicate statistically significantly at p ≤ 0.05 depending on the collagen type and dose; A,B—Mean values denoted by different capital letters indicate statistically significantly at p ≤ 0.05 depending on the day of storage; LA: control milk; LA1.5W: milk with 1.5% collagen; LA3.0W: milk with 3.0% collagen; LA1.5H: milk with 1.5% collagen protein hydrolysate; LA3.0H: milk with 3.0% collagen protein hydrolysate.
Figure 3. Effect of collagen addition on pH parameters of milk fermented by L. paracasei after 1 and 21 days of cold storage.
Mean ± standard deviation; n = 20; a–d—Mean values denoted by different lowercase letters indicate statistically significantly at p ≤ 0.05 depending on the collagen type and dose; A,B—Mean values denoted by different capital letters indicate statistically significantly at p ≤ 0.05 depending on the day of storage; LP: control milk; LP1.5W: milk with 1.5% collagen; LP3.0W: milk with 3.0% collagen; LP1.5H: milk with 1.5% collagen protein hydrolysate; LP3.0H: milk with 3.0% collagen protein hydrolysate.
Figure 4. Effect of collagen addition on pH parameters of milk fermented by L. rhamnosus after 1 and 21 days of cold storage.
Mean ± standard deviation; n = 20; a–d— Mean values denoted by different lowercase letters indicate statistically significantly at p ≤ 0.05 depending on the collagen type and dose; A,B—Mean values denoted by different capital letters indicate statistically significantly at p ≤ 0.05 depending on the day of storage; LR: control milk; LR1.5W: milk with 1.5% collagen; LR3.0W: milk with 3.0% collagen; LR1.5H: milk with 1.5% collagen protein hydrolysate; LR3.0H: milk with 3.0% collagen protein hydrolysate.
Figure 5. Effect of collagen addition on color parameters of milk fermented by L. casei after 1 (A) and 21 (B) days of cold storage.
Mean ± standard deviation; n = 20; a–d— Mean values denoted by different lowercase letters indicate statistically significantly at p ≤ 0.05 depending on the collagen type and dose; A,B—Mean values denoted by different capital letters indicate statistically significantly at p ≤ 0.05 depending on the day of storage; LC: control milk; LC1.5W: milk with 1.5% collagen; LC3.0W: milk with 3.0% collagen; LC1.5H: milk with 1.5% collagen protein hydrolysate; LC3.0H: milk with 3.0% collagen protein hydrolysate.
Figure 6. Effect of collagen addition on color parameters of milk fermented by L. acidophilus after 1 (A) and 21 (B) days of cold storage.
Mean ± standard deviation; n = 20; a–d— Mean values denoted by different lowercase letters indicate statistically significantly at p ≤ 0.05 depending on the collagen type and dose; A,B—Mean values denoted by different capital letters indicate statistically significantly at p ≤ 0.05 depending on the day of storage; LC: control milk; LC1.5W: milk with 1.5% collagen; LC3.0W: milk with 3.0% collagen; LC1.5H: milk with 1.5% collagen protein hydrolysate; LC3.0H: milk with 3.0% collagen protein hydrolysate.
Figure 7. Effect of collagen addition on color parameters of milk fermented by L. paracasei after 1 (A) and 21 (B) days of cold storage.
Mean ± standard deviation; n = 20; a–d— Mean values denoted by different lowercase letters indicate statistically significantly at p ≤ 0.05 depending on the collagen type and dose; A,B—Mean values denoted by different capital letters indicate statistically significantly at p ≤ 0.05 depending on the day of storage; LC: control milk; LC1.5W: milk with 1.5% collagen; LC3.0W: milk with 3.0% collagen; LC1.5H: milk with 1.5% collagen protein hydrolysate; LC3.0H: milk with 3.0% collagen protein hydrolysate.
Figure 8. Effect of collagen addition on color parameters of milk fermented by L. rhamnosus after 1 (A) and 21 (B) days of cold storage.
Mean ± standard deviation; n = 20; a–d— Mean values denoted by different lowercase letters indicate statistically significantly at p ≤ 0.05 depending on the collagen type and dose; A,B—Mean values denoted by different capital letters indicate statistically significantly at p ≤ 0.05 depending on the day of storage; LC: control milk; LC1.5W: milk with 1.5% collagen; LC3.0W: milk with 3.0% collagen; LC1.5H: milk with 1.5% collagen protein hydrolysate; LC3.0H: milk with 3.0% collagen protein hydrolysate.
I suggest adding subtitles to highlight each result with its respective discussion, and avoid confusion.
Subtitles were added:
L93: 2.1. Effect of Pasteurization on pH and Color of Milk with Collagen Addition
L113: 2.2. Acidity of Probiotic Goat’s Milk with Collagen
L261: 2.3. Syneresis of Probiotic Goat’s Milk with Collagen
L297: 2.4. Color of Probiotic Goat’s Milk with Collagen
L362: 2.5. Texture Profile of Probiotic Goat’s Milk with Collagen
L431: 2.6. Viability of Probiotic Bacteria in Goat’s Milk with Collagen
L502: 2.7. Organoleptic Evaluation of Probiotic Goat’s Milk with Collagen

Reviewer 2 Report
In the present work, the authors obtained fermented goat milks with different probiotic strains evaluating the effect of collagen and collagen hydrolysate on the physicochemical properties of the products. Although the topic of the manuscript is interesting and the research design is appropriate, I consider results presentations must be significantly improved in order to ease their reading and comprehension. All the physicochemical parameters of fermented milks obtained with different strains are distributed in four different tables (one table for each strain, Table 2-5), which forces the reader to go up and down along the manuscript to compare results. I suggest the authors present a specific parameter (i.e. pH; lactic acid; % syneresis) in a table/figure comparing the effect of collagen/hydrolysate for all the strains tested. This would significantly simplify the analysis of results and will allow easily evidencing what the authors described in the text. Moreover, I strongly suggest that the authors include some of these results in a figure format instead of table.
There are also some minor and specific corrections that I propose to improve the manuscript:
Lines 47-70: I suggest the authors revised this paragraph in order to organize the information and avoid repeating data.
The authors mentioned several times the relevance of exopolysaccharide production on the texture of fermented milks. However, they did not indicate if any of the strains tested has the ability to produce exopolysaccharide.
Lines 186-187 “LP3.0H and samples with bovine collagen LP3.0W, where it was determined, respectively: 1.35 g L -1 and 1.34 g L -1.” These values are not evidenced in the corresponding Table.
Lines 293-294: “This study showed the best survival rate for L. casei after 21 days in milk with collagen hydrolysate”. In Table 6 it can be observed that no significant differences in bacterial viability between 1 and 21 days of storage were observed for L. rhamnosus, L. paracasei and L. acidophilus. Why do the authors affirm that L. casei is the strain with the best survival rate? In fact, this is the only strain where a significant reduction in viable bacteria was observed after 21 days of storage.
Lines 319: Table 4 and 5? I think the authors refer to Table 6.
Line 416: Please indicate what does TBC and SCC mean between brackets
Lines 464: This tittle should read “Titratable acidity expressed as lactic acid content” instead of “Measurement of Lactic acid concentration” since the authors do not perform a direct measure of lactic acid concentration.
Author Response
Reviewer 2
The authors would like to thank for valuable comments and suggestions. The text of the manuscript has been revised as recommended.
In the present work, the authors obtained fermented goat milks with different probiotic strains evaluating the effect of collagen and collagen hydrolysate on the physicochemical properties of the products. Although the topic of the manuscript is interesting and the research design is appropriate, I consider results presentations must be significantly improved in order to ease their reading and comprehension. All the physicochemical parameters of fermented milks obtained with different strains are distributed in four different tables (one table for each strain, Table 2-5), which forces the reader to go up and down along the manuscript to compare results. I suggest the authors present a specific parameter (i.e. pH; lactic acid; % syneresis) in a table/figure comparing the effect of collagen/hydrolysate for all the strains tested. This would significantly simplify the analysis of results and will allow easily evidencing what the authors described in the text. Moreover, I strongly suggest that the authors include some of these results in a figure format instead of table.
Some of the results (pH value and color of probiotic fermented milk) in Table 3-6 are removed and presented in Figures 1-8:
L161-193: Figures 1-4
L325-360: Figures 5-8
Figure 1. Effect of collagen addition on pH parameters of milk fermented by L. casei after 1 and 21 days of cold storage.
Mean ± standard deviation; n = 20; a–d— Mean values denoted by different lowercase letters indicate statistically significantly at p ≤ 0.05 depending on the collagen type and dose; A,B—Mean values denoted by different capital letters indicate statistically significantly at p ≤ 0.05 depending on the day of storage; LC: control milk; LC1.5W: milk with 1.5% collagen; LC3.0W: milk with 3.0% collagen; LC1.5H: milk with 1.5% collagen protein hydrolysate; LC3.0H: milk with 3.0% collagen protein hydrolysate.
Figure 2. Effect of collagen addition on pH parameters of milk fermented by L. acidophilus after 1 and 21 days of cold storage.
Mean ± standard deviation; n = 20; a–d— Mean values denoted by different lowercase letters indicate statistically significantly at p ≤ 0.05 depending on the collagen type and dose; A,B—Mean values denoted by different capital letters indicate statistically significantly at p ≤ 0.05 depending on the day of storage; LA: control milk; LA1.5W: milk with 1.5% collagen; LA3.0W: milk with 3.0% collagen; LA1.5H: milk with 1.5% collagen protein hydrolysate; LA3.0H: milk with 3.0% collagen protein hydrolysate.
Figure 3. Effect of collagen addition on pH parameters of milk fermented by L. paracasei after 1 and 21 days of cold storage.
Mean ± standard deviation; n = 20; a–d—Mean values denoted by different lowercase letters indicate statistically significantly at p ≤ 0.05 depending on the collagen type and dose; A,B—Mean values denoted by different capital letters indicate statistically significantly at p ≤ 0.05 depending on the day of storage; LP: control milk; LP1.5W: milk with 1.5% collagen; LP3.0W: milk with 3.0% collagen; LP1.5H: milk with 1.5% collagen protein hydrolysate; LP3.0H: milk with 3.0% collagen protein hydrolysate.
Figure 4. Effect of collagen addition on pH parameters of milk fermented by L. rhamnosus after 1 and 21 days of cold storage.
Mean ± standard deviation; n = 20; a–d— Mean values denoted by different lowercase letters indicate statistically significantly at p ≤ 0.05 depending on the collagen type and dose; A,B—Mean values denoted by different capital letters indicate statistically significantly at p ≤ 0.05 depending on the day of storage; LR: control milk; LR1.5W: milk with 1.5% collagen; LR3.0W: milk with 3.0% collagen; LR1.5H: milk with 1.5% collagen protein hydrolysate; LR3.0H: milk with 3.0% collagen protein hydrolysate.
Figure 5. Effect of collagen addition on color parameters of milk fermented by L. casei after 1 (A) and 21 (B) days of cold storage.
Mean ± standard deviation; n = 20; a–d— Mean values denoted by different lowercase letters indicate statistically significantly at p ≤ 0.05 depending on the collagen type and dose; A,B—Mean values denoted by different capital letters indicate statistically significantly at p ≤ 0.05 depending on the day of storage; LC: control milk; LC1.5W: milk with 1.5% collagen; LC3.0W: milk with 3.0% collagen; LC1.5H: milk with 1.5% collagen protein hydrolysate; LC3.0H: milk with 3.0% collagen protein hydrolysate.
Figure 6. Effect of collagen addition on color parameters of milk fermented by L. acidophilus after 1 (A) and 21 (B) days of cold storage.
Mean ± standard deviation; n = 20; a–d— Mean values denoted by different lowercase letters indicate statistically significantly at p ≤ 0.05 depending on the collagen type and dose; A,B—Mean values denoted by different capital letters indicate statistically significantly at p ≤ 0.05 depending on the day of storage; LC: control milk; LC1.5W: milk with 1.5% collagen; LC3.0W: milk with 3.0% collagen; LC1.5H: milk with 1.5% collagen protein hydrolysate; LC3.0H: milk with 3.0% collagen protein hydrolysate.
Figure 7. Effect of collagen addition on color parameters of milk fermented by L. paracasei after 1 (A) and 21 (B) days of cold storage.
Mean ± standard deviation; n = 20; a–d— Mean values denoted by different lowercase letters indicate statistically significantly at p ≤ 0.05 depending on the collagen type and dose; A,B—Mean values denoted by different capital letters indicate statistically significantly at p ≤ 0.05 depending on the day of storage; LC: control milk; LC1.5W: milk with 1.5% collagen; LC3.0W: milk with 3.0% collagen; LC1.5H: milk with 1.5% collagen protein hydrolysate; LC3.0H: milk with 3.0% collagen protein hydrolysate.
Figure 8. Effect of collagen addition on color parameters of milk fermented by L. rhamnosus after 1 (A) and 21 (B) days of cold storage.
Mean ± standard deviation; n = 20; a–d— Mean values denoted by different lowercase letters indicate statistically significantly at p ≤ 0.05 depending on the collagen type and dose; A,B—Mean values denoted by different capital letters indicate statistically significantly at p ≤ 0.05 depending on the day of storage; LC: control milk; LC1.5W: milk with 1.5% collagen; LC3.0W: milk with 3.0% collagen; LC1.5H: milk with 1.5% collagen protein hydrolysate; LC3.0H: milk with 3.0% collagen protein hydrolysate.
There are also some minor and specific corrections that I propose to improve the manuscript:
Lines 47-70: I suggest the authors revised this paragraph in order to organize the information and avoid repeating data.
The paragraph has been checked and corrected (L50-61).
The authors mentioned several times the relevance of exopolysaccharide production on the texture of fermented milks. However, they did not indicate if any of the strains tested has the ability to produce exopolysaccharide.
The discussion was completed with information on the amount of EPS produced by each species of probiotic bacteria based on data from the literature.
Added to the discussion (L371-388):
About 30 species of EPS-producing Lactobacillus have been identified, but the best studied include L. casei, L. acidophilus, L. brevis, L. curvatus, L. delbrueckii ssp. bulgaricus, L. helveticus, L. rhamnosus, L. plantarum and L. johnsonii. In the study by Oleksy and Klewicka [56], EPS production, depending on the type of L. rhamnosus strain and substrate used, ranging from 68 mg L -1 to 137 mg L -1. The L. rhamnosus investigated by Polak-Berecka et al. [57] synthesized EPS ranging from 153 mg L -1 to 185 mg L -1. In contrast, Dupont et al. [58] achieved productivity ten times higher - 1275 mg L -1 EPS produced by L. rhamnosus. As reported by Badel et al. [59], L. rhamnosus is typified as the most efficient producer among Lactobacillus ssp. This strain was reported to synthesize approximately 2.7 g L -1 [59]. Reduction of pH value promotes EPS production efficiency. A similar result was observed for L. casei [60]. In a study by Mozzi et al. [61], EPS production by L. casei ranged from 315 mg L -1 to 488 mg L -1. However, in the study by Kojic et al. [62] produced 140 mg L -1 to 185 mg L -1. The opposite effect of pH impact on the amount of EPS produced was observed for L. acidophilus. In the study by Deepak et al. [63], the production of exopolysaccharides ranged from 215 mg L -1 to 597 mg L -1. In Dupont et al. [58] study, the EPS production by L. paracasei was 85 mg L -1. The production efficiency of EPS depends on the pH value, the type of substrate used, the concentration and source of carbon, temperature, and time [58].
Dupont, I.; Roy, D.; Lapointe, G. Comparison of exopolysaccharide production by strains of Lactobacillus rhamnosus and Lactobacillus paracasei grown in chemically defined medium and milk. J. Ind. Microbiol. Biotechnol. 2000, 24, 251-255.
Lines 186-187 “LP3.0H and samples with bovine collagen LP3.0W, where it was determined, respectively: 1.35 g L -1 and 1.34 g L -1.” These values are not evidenced in the corresponding Table.
The sentence was corrected:
L211-213: After 21 days of storage, the highest lactic acid content was determined in samples fermented by L. casei with a 3.0% addition of hydrolysate LC3.0H and samples with bovine collagen LC3.0W, where it was determined, respectively: 1.35 g L -1 and 1.34 g L -1.
Lines 293-294: “This study showed the best survival rate for L. casei after 21 days in milk with collagen hydrolysate”. In Table 6 it can be observed that no significant differences in bacterial viability between 1 and 21 days of storage were observed for L. rhamnosus, L. paracasei and L. acidophilus. Why do the authors affirm that L. casei is the strain with the best survival rate? In fact, this is the only strain where a significant reduction in viable bacteria was observed after 21 days of storage.
The sentence was corrected:
L477-479: The analysis of the results in Table 7 shows that adding collagen hydrolysate to probiotic milk had no significant effect on the viability of L. rhamnosus and L. paracasei on day 1 and 21 of storage.
The sentence “This study showed the best survival rate for L. casei after 21 days in milk with collagen hydrolysate” was changed:
L449-451: The best survival rate after 21 days of cold storage compared to day one was found in milk with 3.0% added collagen hydrolysate fermented by L. casei (LC3.0H). The survival rate was 103.66%.
Lines 319: Table 4 and 5? I think the authors refer to Table 6.
The Table number was changed to “Table 7” (L77-479).
Line 416: Please indicate what does TBC and SCC mean between brackets
L576-577: The meaning of the acronymus TBC and SCC was explained:
“IBCm Bacto Kit 500 and IBCm SCC Kit reagents for determining TBC (total bacterial count) and SCC (somatic cell count) were purchased from Bentley Instruments Inc. (Chaska, MN, USA).”
Lines 464: This tittle should read “Titratable acidity expressed as lactic acid content” instead of “Measurement of Lactic acid concentration” since the authors do not perform a direct measure of lactic acid concentration
The title has been changed as was suggested. Now is: Titratable acidity expressed as lactic acid content (L620).

Reviewer 3 Report
The topic of the paper “Effect of Storage Time and Bacterial Strain on the Quality of Probiotic Goat’s Milk with Different Types of Collagens” results to be interesting; the background and the objectives of the work are well described, and the experimental results collected for the product characterization are complete and exhaustive. Mainly, Materials and Methods section needs to be improved, as well as a more schematic description of the experimental plan would be preferable. Moreover, discussion of results should be expanded. For these reasons, I suggest applying some revisions before publication in Molecules.

Author Response
Reviewer 3
The authors would like to thank for valuable comments and suggestions. The text of the manuscript has been revised as recommended.
The topic of the paper “Effect of Storage Time and Bacterial Strain on the Quality of Probiotic Goat’s Milk with Different Types of Collagens” results to be very interesting; the background and the objectives of the work are well described, and the experimental results collected for the product characterization are complete and exhaustive. Mainly, Materials and Methods section needs to be improved, as well as a more schematic description of the experimental plan would be preferable. Moreover, discussion of results should be expanded. For these reasons, I suggest applying some revisions before publication in Molecules.
Title: I suggest modifying the title as “Effect of Storage Time and bacterial Strain on the Quality of Probiotic Goat’s Milk using different types and doses of Collagens”, to better summarize the objectives of the paper.
L2-4: The title has been changed as was suggested.
Results and Discussion
Lines 95-104: Considerations related to the possibility of using collagen to improve the texture of fermented goat’s milks would be more appropriate to be explained in Introduction than in Results. In any case, they can be only briefly mentioned at the beginning of this section or, even better, later in the text (when results on texture properties are shown and discussed).
Lines 95-100 were moved to chapter 1. Introduction (L57-61).
Lines 100-104 were moved to chapter 2.5. Texture Profile of Probiotic Goat’s Milk with Collagen (L394-40).
Line 111: How did you verify the absence of protein denaturation after pasteurization at 85°C for 30 min? Moreover, isn’t this heat treatment too extreme to be a pasteurization?
The heat stability of milk is an indicator of milk's protein stability and an essential technological property of raw milk. The heat stability of milk is its ability to withstand a defined heat treatment without noticeable changes, such as the flocculation of protein. Milk stability is considered the total time for visual coagulation to occur at a given pH and temperature, and it is directly related to the ability of milk to resist coagulation at specific temperatures. To determine heat stability, a simple technological test is usually used. Changes in the pH of milk or the addition of collagen may adversely affect heat stability. The study aimed to determine the processing suitability of goat milk with collagen added to the milk before pasteurization for the production of probiotic fermented milk. Therefore, the authors decided to conduct this simple test to see how individual doses of collagen would affect the pH value of the milk after pasteurization (Table 2) and whether there would be visible coagulation of the milk, possibly noticeable changes, such as flocculation of proteins. This simple method, which involved measuring the pH value and observing visible changes, was intended to answer whether milk with a particular dose or type of collagen would successfully pass the pasteurization process and could be used to make probiotic fermented milk.
The recommended pasteurization temperature for yogurt or fermented milk production is 90 to 95°C for close to a minute to a few minutes (HTST) or 85°C for 30 minutes [1]. The high temperature also denatures whey proteins, allowing the yogurt to form a more stable gel. In our study raw goat’s milk was pasteurized at 85 °C, for 30 minutes. After adding different doses and types of collagen, the milk was re-pasteurized at 85°C, for 10 minutes. There was a mistake in Line 111, which was corrected according to the fermented milk production method, i.e. 85°C, 10 minutes (now in Line 595).
- Aryana, K.J.; Olson, D.W. A 100-Year Review: Yogurt and other cultured dairy products. Journal of Dairy Science 2017, 100 (12), 9987 – 10013. DOI:https://doi.org/10.3168/jds.2017-12981
Line 123-124: Please, clarify better the meaning of this phrase.
L116-125: Some authors indicated that the higher fermentation activity of lactic acid bacteria in goat's milk is due to its specific composition and structure [35]. The buffering capacity of goat's milk is lower than cow's and sheep's milk. During the manufacture of yogurt from goat's milk, a pH of 4.6 to 4.7 is achieved after 2 h 45 min, in cow's milk - after 3 h 30 min, and in sheep's milk - after 5 h 30 min. The faster pH changes in goat's milk are mainly due to its lower casein content, while β-casein is the least phosphorylated. Moreover, the higher non-protein nitrogen content, higher vitamin concentration, higher amount of some minerals and short-chain fatty acids, and easier protein digestibility in goat's milk may affect the faster increase in acidity [36].
Lines130-131: Please, discuss more the reasons why collagen addition increases pH both before fermentation (1st day of storage) and after 21 days of storage, and the differences in pH between hydrolysate and bovine collagen samples.
Added to the discussion (L136-154):
It was found that the addition of 3.0% hydrolysate and bovine collagen results in an increase in pH values by 0.42 in LA3.0W samples and 0.54 in LA3.0H samples of fermented milk. In the case of our study, collagen was added to milk with a pH value of 6.8. Under these conditions, at pH values below isoelectric point, there is an increase in hydrogen ions and protein particles are positively charged, causing repulsion between them [37,38]. Therefore, the pH value is lower in milk with collagen than in control milk. Also in Goto's [39] study, the addition of 1.5% bovine collagen resulted in a 0.14 reduction in pH values, and the addition of 3.0% collagen resulted in a 0.23 decrease in pH values.
After 21 days of storage, all samples with collagen showed higher pH values than their control counterparts. In fermented milk, after 21 days of storage, adding 1.5%, bovine collagen increased the pH value from 0.03 in LR1.5W to 0.17 in LA1.5W. However, adding 3% collagen increased: 0.17 in LR3.0W and 0.31 in LA3.0W.After 21 days of storage, the addition of 1.5% hydrolysate increased the pH value compared to controls from 0.11 in LP1.5H to 0.28 in LA1.5H. Increasing the dose resulted in higher pH values, by 0.20 in LP3.0H and LC3.0H to 0.39 in LA3.0H. Studies by Goto [39] indicated a slow reduction in pH in milk containing 3.06% and 5.1% collagen, indicating a delayed fermentation phenomenon caused by the addition of collagen. Similar results were obtained in milk with different types of collagen, i.e. fish and pork [39].
Line 176-189: Please, discuss more about the differences in lactic acid concentration obtained with and without addition of two different types of collagens and by using different strains.
Added to the discussion (L201-226):
On day 1 of storage of samples fermented by L. paracasei, the addition of bovine collagen was found to increase the lactic acid content by 0.1 g L -1 compared to the control sample while increasing the dose of bovine collagen from 1.5% to 3.0% did not significantly affect the lactic acid content of fermented milk. However, the hydrolysate addition to milk fermented by L. paracasei increased lactic acid content by 0.08 but only at the 3.0% hydrolysate dose. At a dose of 1.5% hydrolysate, the lactic acid content was the same as in the control sample on day 1 of storage (Table 5).
The highest concentration of lactic acid after 21 days of storage was found in goat's milk fermented by L. casei. In this case, adding bovine collagen at 1.5% and 3.0% increased the acid content by 0.04 and 0.12 g L -1, respectively. Also, the addition of hydrolysate increased the lactic acid concentration in the LC1.5H sample by 0.08 and the LC3.0H sample by 0.13 after 21 days of storage. Moreover, extending the storage time to 21 days resulted in the highest increase in lactic acid concentration in samples fermented by L. casei compared to the other fermented milk samples.
In a study performed by Shori et al. [42] the addition of fish collagen increased the initial titratable acidity (TA%) by about 0.2% lactic acid equivalent. In yogurts with fish collagen, there was an increase in TTA (total titratable acidity) throughout refrigerated storage compared to the control sample[43].
Sometimes, in the text it is not clear if the discussed comparisons related to pH, lactic acid, texture, and syneresis are referred to fermented milks at 1st day of storage or to fermented milks after 21 days. Please, rearrange discussions with major clarity.
Moreover, a subdivision in paragraphs of this section could be useful.
The discussion was reviewed and divided. Paragraphs and subsections were added.
Subtitles were added:
L93: 2.1. Effect of Pasteurization on pH and Color of Milk with Collagen Addition
L113: 2.2. Acidity of Probiotic Goat’s Milk with Collagen
L261: 2.3. Syneresis of Probiotic Goat’s Milk with Collagen
L297: 2.4. Color of Probiotic Goat’s Milk with Collagen
L362: 2.5. Texture Profile of Probiotic Goat’s Milk with Collagen
L431: 2.6. Viability of Probiotic Bacteria in Goat’s Milk with Collagen
L502: 2.7. Organoleptic Evaluation of Probiotic Goat’s Milk with Collagen
Lines 263-264: Was the organic acids and exopolysaccharides production during fermentation confirmed and quantified in this work?
In the study, the determination of total acidity expressed as lactic acid content was carried out. Exopolysaccharides (EPS) and individual organic acids produced by probiotic bacteria were not measured quantitatively. The ability to produce EPS is due to the properties of the individual strains, which has been confirmed in studies by other authors. The discussion was completed with information on the amount of EPS produced by each species of probiotic bacteria based on data from the literature.
Added to the discussion (L371-388):
About 30 species of EPS-producing Lactobacillus have been identified, but the best studied include L. casei, L. acidophilus, L. brevis, L. curvatus, L. delbrueckii ssp. bulgaricus, L. helveticus, L. rhamnosus, L. plantarum and L. johnsonii. In the study by Oleksy and Klewicka [56], EPS production, depending on the type of L. rhamnosus strain and substrate used, ranging from 68 mg L -1 to 137 mg L -1. The L. rhamnosus investigated by Polak-Berecka et al. [57] synthesized EPS ranging from 153 mg L -1 to 185 mg L -1. In contrast, Dupont et al. [58] achieved productivity ten times higher - 1275 mg L -1 EPS produced by L. rhamnosus. As reported by Badel et al. [59], L. rhamnosus is typified as the most efficient producer among Lactobacillus ssp. This strain was reported to synthesize approximately 2.7 g L -1 [59]. Reduction of pH value promotes EPS production efficiency. A similar result was observed for L. casei [60]. In a study by Mozzi et al. [61], EPS production by L. casei ranged from 315 mg L -1 to 488 mg L -1. However, in the study by Kojic et al. [62] produced 140 mg L -1 to 185 mg L -1. The opposite effect of pH impact on the amount of EPS produced was observed for L. acidophilus. In the study by Deepak et al. [63], the production of exopolysaccharides ranged from 215 mg L -1 to 597 mg L -1. In Dupont et al. [58] study, the EPS production by L. paracasei was 85 mg L -1. The production efficiency of EPS depends on the pH value, the type of substrate used, the concentration and source of carbon, temperature, and time [58].
Dupont, I.; Roy, D.; Lapointe, G. Comparison of exopolysaccharide production by strains of Lactobacillus rhamnosus and Lactobacillus paracasei grown in chemically defined medium and milk. J. Ind. Microbiol. Biotechnol. 2000, 24, 251-255.
Lines 273-274: Please, discuss with some hypothesis the cohesiveness reduction observed with 1.5% bovine collagen addition.
Added to the discussion (L416-428):
However, statistically, significant differences were shown only for LP1.5W samples after 21 days of storage. Also, in the Luo et al. [73] study, gel cohesiveness was not affected by the addition of 0.1% or 0.2% gelatin but increased significantly at higher gelatin concentrations. In a study by Pang et al. [74], the cohesiveness of milk gel samples also decreased significantly as the gelatin concentration increased to 1%, which could be related to steric interference by gelatin addition. These authors, studying the microstructure, observed that low concentrations of gelatin did not significantly change the gel matrix of milk [74]. An analysis of variance confirmed that the texture components (hardness, cohesiveness, springiness) were most influenced by the type of probiotic strain used, the type of collagen, and the interaction of these two factors. In a study by Szopa et al. [9] in sheep's milk fermented by L. casei, a reduction in milk cohesiveness was observed in samples with 1.5 and 3.0% collagen hydrolysate addition compared to the control sample throughout the storage period.
Line 319: Why did you mention Tables 4 and to talk about viability of L. rhamnosus and L.paracasei?
The sentence was corrected:
L477-479: The analysis of the results in Table 7 shows that adding collagen hydrolysate to probiotic milk had no significant effect on the viability of L. rhamnosus and L. paracasei on day 1 and 21 of storage.
Table 6: to introduce an additional column for the survival percentage could be useful.
A line was added to the methodology in section 3.3.7.:
L663-664: From the obtained results, the percentage survival rate of probiotic bacteria was calculated.
L457-475: Table 7. Viable counts of probiotic bacteria [log cfu g -1] in fermented goat’s milk and the survival percentage of probiotic bacteria.
|
Fermented milk group |
Storage Time (days) |
The survival (%) |
|
|
1 |
21 |
||
|
LR |
8.73 aA ± 0.43 |
8.67 aA ± 0.29 |
99.31 |
|
LR1.5W |
9.31 bA ± 0.72 |
8.82 aA ± 0.35 |
94.75 |
|
LR3.0W |
9.28 bA ± 0.04 |
8.64 aA ± 0.19 |
93.10 |
|
LR1.5H |
8.90 aA ± 0.40 |
8.88 aA ± 0.71 |
99.78 |
|
LR3.0H |
9.17 abA ± 0.11 |
8.66 aA ± 0.07 |
94.44 |
|
LP |
8.93 aA ± 0.36 |
8.83 aA ± 0.11 |
98.88 |
|
LP1.5W |
9.15 aA ± 0.66 |
8.85 aA ± 0.22 |
96.72 |
|
LP3.0W |
8.99 aA ± 0.71 |
8.93 aA ± 0.29 |
99.33 |
|
LP1.5H |
8.97 aA ± 0.45 |
8.85 aA ± 0.20 |
98.66 |
|
LP3.0H |
9.18 aA ± 0.12 |
8.93 aA ± 0.65 |
97.28 |
|
LA |
8.97 aB ± 0.18 |
8.24 aA ± 0.32 |
91.86 |
|
LA1.5W |
8.89 aA ± 0.38 |
8.58 bA ± 0.11 |
96.51 |
|
LA3.0W |
8.83 aA ± 0.50 |
8.68 abA ± 0.74 |
98.30 |
|
LA1.5H |
8.96 aA ± 0.22 |
8.41 abA ± 0.86 |
93.86 |
|
LA3.0H |
8.91 aA ± 0.30 |
8.63 bA ± 0.17 |
96.86 |
|
LC |
9.24 aB ± 0.29 |
8.29 aA ± 0.12 |
89.72 |
|
LC1.5W |
9.29 aB ± 0.13 |
8.36 aA ± 0.58 |
89.99 |
|
LC3.0W |
9.05 aB ± 0.65 |
8.54 abA ± 0.12 |
94.37 |
|
LC1.5H |
8.98 aB ± 0.12 |
8.68 bA ± 0.11 |
96.66 |
|
LC3.0H |
9.01 aB ± 0.30 |
9.34 bA ± 0.27 |
103.66 |
A-B — mean values denoted for one probiotic strain in storage time by different letters differ significantly at p ≤ 0.05; a-b — mean values denoted for one probiotic strain in collagen type and dose for given different letters differ significantly at p ≤ 0.05. Storage time: 1 - after fermentation; 21 - after 21 days; LP: control milk with Lacticaseibacillus paracasei; LP1.5W: milk with 1.5% collagen and Lacticaseibacillus paracasei; LP3.0W: milk with 3.0% collagen and Lacticaseibacillus paracasei; LP1.5H: milk with 1.5% collagen hydrolysate and Lacticaseibacillus paracasei; LP3.0H: milk with 3.0% collagen hydrolysate and Lacticaseibacillus paracasei; LR: control milk with Lacticaseibacillus rhamnosus; LR1.5W: milk with 1.5% collagen and Lacticaseibacillus rhamnosus; LR3.0W: milk with 3.0% collagen and Lacticaseibacillus rhamnosus; LR1.5H: milk with 1.5% collagen hydrolysate and Lacticaseibacillus rhamnosus; LR3.0H: milk with 3.0% collagen hydrolysate and Lacticaseibacillus rhamnosus.; ; LC: control milk with Lacticaseibacillus casei; LC1.5W: milk with 1.5% collagen and Lacticaseibacillus casei; LC3.0W: milk with 3.0% collagen and Lacticaseibacillus casei; LC1.5H: milk with 1.5% collagen hydrolysate and Lacticaseibacillus casei; LC3.0H: milk with 3.0% collagen hydrolysate and Lacticaseibacillus casei; LA: control milk with Lactobacillus acidophilus; LA1.5W: milk with 1.5% collagen and Lactobacillus acidophilus; LA3.0W: milk with 3.0% collagen and Lactobacillus acidophilus; LA1.5H: milk with 1.5% collagen hydrolysate and Lactobacillus acidophilus; LA3.0H: milk with 3.0% collagen hydrolysate and Lactobacillus acidophilus.
Moreover, more references to the figures as the results are presented and discussed may be helpful to the reader.
References to the Figures 1-8 are included in the manuscript's text.
Some of the results (pH value and color of probiotic fermented milk) in Table 3-6 are removed and presented in Figures 1-8:
L161-193: Figures 1-4
L325-360: Figures 5-8
Figure 1. Effect of collagen addition on pH parameters of milk fermented by L. casei after 1 and 21 days of cold storage.
Mean ± standard deviation; n = 20; a–d— Mean values denoted by different lowercase letters indicate statistically significantly at p ≤ 0.05 depending on the collagen type and dose; A,B—Mean values denoted by different capital letters indicate statistically significantly at p ≤ 0.05 depending on the day of storage; LC: control milk; LC1.5W: milk with 1.5% collagen; LC3.0W: milk with 3.0% collagen; LC1.5H: milk with 1.5% collagen protein hydrolysate; LC3.0H: milk with 3.0% collagen protein hydrolysate.
Figure 2. Effect of collagen addition on pH parameters of milk fermented by L. acidophilus after 1 and 21 days of cold storage.
Mean ± standard deviation; n = 20; a–d— Mean values denoted by different lowercase letters indicate statistically significantly at p ≤ 0.05 depending on the collagen type and dose; A,B—Mean values denoted by different capital letters indicate statistically significantly at p ≤ 0.05 depending on the day of storage; LA: control milk; LA1.5W: milk with 1.5% collagen; LA3.0W: milk with 3.0% collagen; LA1.5H: milk with 1.5% collagen protein hydrolysate; LA3.0H: milk with 3.0% collagen protein hydrolysate.
Figure 3. Effect of collagen addition on pH parameters of milk fermented by L. paracasei after 1 and 21 days of cold storage.
Mean ± standard deviation; n = 20; a–d—Mean values denoted by different lowercase letters indicate statistically significantly at p ≤ 0.05 depending on the collagen type and dose; A,B—Mean values denoted by different capital letters indicate statistically significantly at p ≤ 0.05 depending on the day of storage; LP: control milk; LP1.5W: milk with 1.5% collagen; LP3.0W: milk with 3.0% collagen; LP1.5H: milk with 1.5% collagen protein hydrolysate; LP3.0H: milk with 3.0% collagen protein hydrolysate.
Figure 4. Effect of collagen addition on pH parameters of milk fermented by L. rhamnosus after 1 and 21 days of cold storage.
Mean ± standard deviation; n = 20; a–d— Mean values denoted by different lowercase letters indicate statistically significantly at p ≤ 0.05 depending on the collagen type and dose; A,B—Mean values denoted by different capital letters indicate statistically significantly at p ≤ 0.05 depending on the day of storage; LR: control milk; LR1.5W: milk with 1.5% collagen; LR3.0W: milk with 3.0% collagen; LR1.5H: milk with 1.5% collagen protein hydrolysate; LR3.0H: milk with 3.0% collagen protein hydrolysate.
Figure 5. Effect of collagen addition on color parameters of milk fermented by L. casei after 1 (A) and 21 (B) days of cold storage.
Mean ± standard deviation; n = 20; a–d— Mean values denoted by different lowercase letters indicate statistically significantly at p ≤ 0.05 depending on the collagen type and dose; A,B—Mean values denoted by different capital letters indicate statistically significantly at p ≤ 0.05 depending on the day of storage; LC: control milk; LC1.5W: milk with 1.5% collagen; LC3.0W: milk with 3.0% collagen; LC1.5H: milk with 1.5% collagen protein hydrolysate; LC3.0H: milk with 3.0% collagen protein hydrolysate.
Figure 6. Effect of collagen addition on color parameters of milk fermented by L. acidophilus after 1 (A) and 21 (B) days of cold storage.
Mean ± standard deviation; n = 20; a–d— Mean values denoted by different lowercase letters indicate statistically significantly at p ≤ 0.05 depending on the collagen type and dose; A,B—Mean values denoted by different capital letters indicate statistically significantly at p ≤ 0.05 depending on the day of storage; LC: control milk; LC1.5W: milk with 1.5% collagen; LC3.0W: milk with 3.0% collagen; LC1.5H: milk with 1.5% collagen protein hydrolysate; LC3.0H: milk with 3.0% collagen protein hydrolysate.
Figure 7. Effect of collagen addition on color parameters of milk fermented by L. paracasei after 1 (A) and 21 (B) days of cold storage.
Mean ± standard deviation; n = 20; a–d— Mean values denoted by different lowercase letters indicate statistically significantly at p ≤ 0.05 depending on the collagen type and dose; A,B—Mean values denoted by different capital letters indicate statistically significantly at p ≤ 0.05 depending on the day of storage; LC: control milk; LC1.5W: milk with 1.5% collagen; LC3.0W: milk with 3.0% collagen; LC1.5H: milk with 1.5% collagen protein hydrolysate; LC3.0H: milk with 3.0% collagen protein hydrolysate.
Figure 8. Effect of collagen addition on color parameters of milk fermented by L. rhamnosus after 1 (A) and 21 (B) days of cold storage.
Mean ± standard deviation; n = 20; a–d— Mean values denoted by different lowercase letters indicate statistically significantly at p ≤ 0.05 depending on the collagen type and dose; A,B—Mean values denoted by different capital letters indicate statistically significantly at p ≤ 0.05 depending on the day of storage; LC: control milk; LC1.5W: milk with 1.5% collagen; LC3.0W: milk with 3.0% collagen; LC1.5H: milk with 1.5% collagen protein hydrolysate; LC3.0H: milk with 3.0% collagen protein hydrolysate.
Materials and methods
Lines 407-408: Please, briefly specify in the text the physicochemical properties used to characterize raw goat milk, reported in Table 7. The composition of raw milk, evaluated by the methods described in section 3.3 can be shown as a preliminary result. Thus, I suggest moving Table 7 to Results section.
Table 7 (now Table 1) has been moved to the "Result and Discussion" section, L84. Moreover, results describing the quality of raw goat's milk were briefly described:
L85-91: Variability in the composition of goat's milk affects the technological and sensory properties [30]. In order to determine the processing suitability of raw goat's milk, the composition and physicochemical characteristics of goat's milk were evaluated (Table 1). The obtained results are comparable with the findings of other studies [16,31,32].
Table 1. Composition and physicochemical properties of raw goat's milk.
|
Properties |
Mean ± SD1 |
|
|
Total solids, g 100 g−1 |
11.45 ± 1.46 |
|
|
Protein, g 100 g−1 |
2.64 ± 0.27 |
|
|
Fat, g 100 g−1 |
3.43 ± 0.66 |
|
|
Lactose, g 100 g−1 |
4.35 ± 0.52 |
|
|
Density, g mL−1 |
1.027 ± 0.003 |
|
|
Freezing point, °C |
-0.603 ± 0.06 |
|
|
pH |
6.88 ± 0.03 |
|
|
Color |
L* |
89.88 ± 1.05 |
|
a* |
-1,95 ± 0.22 |
|
|
b* |
8.13 ± 1.01 |
|
|
C |
8.21 ± 1.03 |
|
|
h° |
97.87 ± 1.61 |
|
|
TBC2, log CFU mL−1 |
6.56 ± 0.21 |
|
|
SCC3, log cells mL−1 |
5.77 ± 0.41 |
|
1 SD—standard deviation; 2 TBC—total bacterial count; 3 SCC—somatic cell count.
Table 8: After a first reading, table 8 appears difficult to understand, so it would be preferable to structure it differently.
L599-600: Table 8 has been removed. A Scheme 1 was prepared, in which the plan of the experiment was described.
|
|||||||||||||||||
|
|||||||||||||||||
Scheme 1. Production scheme for probiotic fermented milk with bovine collagen and collagen hydrolysate.
Sections 3.3 and 3.4: It would be preferable not to repeat the analytical methods (as for pH, or colour) used for both raw and fermented milk in each paragraph. It is possible to just have a single paragraph (i.e., section 3.3 “analytical methods” or “analysis of raw and fermented goat’s milk”, as you prefer) containing all the analysis and the analytical methods used, subdivided in subparagraphs (as 3.3.1, 3.3.2, etc..).
One section was created: 3.3. Methods of Analyses containing all the analysis and the analytical methods used:
L609: 3.3. Methods of Analyses
L610: 3.3.1. Determination of Freezing Point and Chemical Composition of Raw Goat’s Milk
L616: 3.3.2. Microbiological and Cytological Quality of Raw Goat’s Milk
L620: 3.3.3. Titratable Acidity Expressed as Lactic Acid Content and pH
L629: 3.3.4. Color Evaluation
L638: 3.3.5. Syneresis
L646: 3.3.6. Texture Profile Analysis
L655: 3.3.7. Microbiological Analysis
L666: 3.3.8. Organoleptic Evaluation
L678: 3.3.9. Statistical Analysis
Moreover, after Materials section and before Analysis section, a paragraph for describing the experimental plan could help.
A scheme (Scheme 1) for the production of fermented milk beverages was prepared, which is also the plan of the experiment; L599-600.
Section 3.4.3. lines 485-487: Maybe it would be better to write that the syneresis percentage was expressed as the amount of whey released compared to the initial sample weight and multiplied for 100.
The sentence was corrected:
L642-644: Syneresis percentage was expressed as the amount of whey released compared to the initial sample weight and multiplied for 100.
Conclusions
In addition to a summary of the main contents, conclusions should also highlight to the reader the findings importance, their implications, and future purposes. Please, review this section in this way.
The conclusions are discussed in more detail:
L709-722:
Overall, our results indicate that probiotic goat milk with collagen has high commercial potential due to its nutritional and functional composition. The supplementation of fermented goat's milk with collagen could be a new direction for broadening the assortment of goat products. However, it still requires improvements in technological parameters - mainly fermentation with different probiotic strains.
Nevertheless, further research is necessary to adequately clarify the impact and correlation between the content of different types of collagen and hydrolysates (not just bovine) and the quality of fermented milk made from the milk of different animal species. In particular, research is required on the survival rate of different strains of probiotic bacteria delivered to the human body in the form of fermented milk with collagen determined by simulated in vitro digestion. Moreover, the release rate of nutrients could also be studied, which is essential for products intended for consumers with special needs.

Round 2
Reviewer 1 Report
The authors correctly addressed the changes in the manuscript, so it can be accepted.
Author Response
The authors would like to thank for the acceptance of the manuscript and positive review.

Reviewer 2 Report
The manuscript has been significantly improved. However, some details must be modified and correct before publication.
Line 137-141: “In the case of our study, collagen was added to milk with a pH value of 6.8. Under these conditions, at pH values below isoelectric point, there is an increase in hydrogen ions and protein particles are positively charged, causing repulsion between them [37,38]. Therefore, the pH value is lower in milk with collagen than in control milk.” I understand the authors are explaining the lower pH evidenced in milk when added with collagen and hydrolysate (before fermentation). Thus, I consider this should be included in the discussion of Section 2.1 Effect of Pasteurization on pH and Color of Milk with Collagen Addition rather than in Section 2.2. Acidity of Probiotic Goat’s Milk with Collagen.
Figure 1: Bar corresponding to LC3.0H 21 days. According to the bar, pH value must be between 4.3 and 4.4 Please, correct the number 4.69.
Line 270-273: “In our study, on day 1 of storage, in most cases, adding a hydrolysate at 3.0% increased syneresis from 0.2% to 9.0%, depending on the type of strain used. Only LA3.0H fermented milk showed a 2.42% reduction in syneresis compared to the control (Table 4).”. If I am correct, there is only a significant increase in syneresis in the case of LP3.0H. LC3.0H and LR3.0H showed no significant differences with the corresponding control while LA3.0H tended to reduce syneresis but differences are not significant with the control.
Lines 274-275: “After 21 days of storage, all samples of milk fermented by L. rhamnosus showed a reduction in syneresis of 1.83-4.53% compared to day 1 of storage”. According to Table 6, after 21 days of storage all L rhamnosus fermented milks showed an INCREASE in syneresis, not a reduction.
Lines 297-299: “In our study, adding hydrolysate and bovine collagen caused color darkening after pasteurization (Table 2) and during 21 days of refrigerated storage (Figure 5-8)”. In Table 2 no significant differences are evidenced when adding hydrolysate and bovine collagen in color darkening as the authors explain in lines 102-103: “Most importantly, the addition of collagen did not significantly change the color components of goat’s milk”.
Lines 301-302: “It was also found that L* color lightness decreased with increasing storage time in all samples”. I would replace it by “L* color lightness TENDS TO decreased”, since differences are significant only in a few cases.
Lines 309-311: “In the case of milk fermented by L. casei, L. acidophilus and L. paracasei (Figure 5-7), there was a reduction in the intensity of the yellow color in the milk on day 1 of storage”. A reduction in yellow with respect to….? Please, specify.
Lines 370-387: This paragraph can be removed. The ability to produce EPS is strain dependent. So, I asked if any of the specific strains used in the present work (Lacticaseibacillus casei 431®, Lacto-571 bacillus acidophilus LA- 5®, Lacticaseibacillus paracasei 572 LP26 and Lacticaseibicillus rhamnosus Lr- 32) produces EPS, in order to analyze EPS production with syneresis and texture parameters and compared the results obtained with different strains tested.
Author Response
Reviewer 2
The authors would like to thank for reviewing the manuscript. The text of the manuscript has been revised as recommended.
The manuscript has been significantly improved. However, some details must be modified and correct before publication.
Line 137-141: “In the case of our study, collagen was added to milk with a pH value of 6.8. Under these conditions, at pH values below isoelectric point, there is an increase in hydrogen ions and protein particles are positively charged, causing repulsion between them [37,38]. Therefore, the pH value is lower in milk with collagen than in control milk.” I understand the authors are explaining the lower pH evidenced in milk when added with collagen and hydrolysate (before fermentation). Thus, I consider this should be included in the discussion of Section 2.1 Effect of Pasteurization on pH and Color of Milk with Collagen Addition rather than in Section 2.2. Acidity of Probiotic Goat’s Milk with Collagen.
Lines 137-141 were moved in the discussion of Section 2.1 Effect of Pasteurization on pH and Color of Milk with Collagen Addition, now L101-104.
Figure 1: Bar corresponding to LC3.0H 21 days. According to the bar, pH value must be between 4.3 and 4.4 Please, correct the number 4.69.
The pH value in the graph for LC 3.0H 21 days has been corrected (L160).
Line 270-273: “In our study, on day 1 of storage, in most cases, adding a hydrolysate at 3.0% increased syneresis from 0.2% to 9.0%, depending on the type of strain used. Only LA3.0H fermented milk showed a 2.42% reduction in syneresis compared to the control (Table 4).”. If I am correct, there is only a significant increase in syneresis in the case of LP3.0H. LC3.0H and LR3.0H showed no significant differences with the corresponding control while LA3.0H tended to reduce syneresis but differences are not significant with the control.
The paragraph was corrected:
L270-276: In our study, on day 1 of storage, in most cases, adding a hydrolysate at 3.0% increased syneresis from 0.2% to 9.0%, depending on the type of strain used (Table 3-6). LA3.0H fermented milk showed a 2.42% reduction in syneresis compared to the control, however, the reported difference was not statistically significant (Table 4). Conversely, the most significant reduction in syneresis compared to the control sample (by 2.58%) was found in milk fermented by L. rhamnosus with the addition of 1.5% hydrolysate (Table 6).
Lines 274-275: “After 21 days of storage, all samples of milk fermented by L. rhamnosus showed a reduction in syneresis of 1.83-4.53% compared to day 1 of storage”. According to Table 6, after 21 days of storage all L rhamnosus fermented milks showed an INCREASE in syneresis, not a reduction.
The sentence was corrected:
L277-278: After 21 days of storage, all samples of milk fermented by L. rhamnosus showed an increase in syneresis of 1.83-4.53% compared to day 1 of storage.
Lines 297-299: “In our study, adding hydrolysate and bovine collagen caused color darkening after pasteurization (Table 2) and during 21 days of refrigerated storage (Figure 5-8)”. In Table 2 no significant differences are evidenced when adding hydrolysate and bovine collagen in color darkening as the authors explain in lines 102-103: “Most importantly, the addition of collagen did not significantly change the color components of goat’s milk”.
The part in the sentence about the color change of milk with collagen after pasteurization was removed.
L300-302: Many studies reported that milk additives affect color parameters. In our study, adding hydrolysate and bovine collagen caused color darkening during 21 days of refrigerated storage (Figure 5-8).
Lines 301-302: “It was also found that L* color lightness decreased with increasing storage time in all samples”. I would replace it by “L* color lightness TENDS TO decreased”, since differences are significant only in a few cases.
The sentence has been revised as suggested:
L304-305: It was also found that L* color lightness tends to decrease with increasing storage time in all samples.
Lines 309-311: “In the case of milk fermented by L. casei, L. acidophilus and L. paracasei (Figure 5-7), there was a reduction in the intensity of the yellow color in the milk on day 1 of storage”. A reduction in yellow with respect to….? Please, specify.
The sentence has been revised as:
L312-318: There was observed a reduction in the intensity of yellow color in milk fermented by L. acidophilus and L. paracasei (Figure 6-7) on day 1 of storage with the addition of both collagen and hydrolysate and in milk with a 3% addition of collagen hydrolysate fermented by L. rhamnosus (Figure 8) compared to control milk. In milk fermented by L. casei, the addition of collagen and hydrolysate in two doses increased the proportion of yellow color on day one as well as day 21 of refrigerated storage (Figure 5).
Lines 370-387: This paragraph can be removed. The ability to produce EPS is strain dependent. So, I asked if any of the specific strains used in the present work (Lacticaseibacillus casei 431®, Lacto-571 bacillus acidophilus LA- 5®, Lacticaseibacillus paracasei 572 LP26 and Lacticaseibicillus rhamnosus Lr- 32) produces EPS, in order to analyze EPS production with syneresis and texture parameters and compared the results obtained with different strains tested.
Paragraph 370-387 has been deleted as suggested.

Reviewer 3 Report
After applying the following suggestions, the paper can be accepted for publication.
In section 3.3.7, it would be preferable to specify how the survival was calculated.
Concerning the first pasteurisation on raw goat's milk, add the reference reported into the response (Aryana and Olson, 2017) to justify the temperature/time conditions adopted.
Author Response
Reviewer 3
The authors would like to thank for reviewing the manuscript. The text of the manuscript has been revised as recommended.
In section 3.3.7, it would be preferable to specify how the survival was calculated.
It was specified how the survival rate of probiotic bacteria was calculated (L652-655):
The survival rate of probiotic bacteria (%) was determined by the number of viable colonies of probiotic bacteria in the fermented milk on day 21 of storage relative to the sample from day 1 of storage, according to Equation 1:
Survival rate of probiotic bacteria (%) = × 100
Concerning the first pasteurisation on raw goat's milk, add the reference reported into the response (Aryana and Olson, 2017) to justify the temperature/time conditions adopted.
The reference reported in the response was added to the manuscript text, L572:
Raw goat’s milk was pasteurized at 85 °C for 30 min [79].
L914-915: 79. Aryana, K.J.; Olson, D.W. A 100-Year Review: Yogurt and other cultured dairy products. Journal of Dairy Science 2017, 100(12), 9987 – 10013. DOI:https://doi.org/10.3168/jds.2017-12981
